# Effects of Vehicle Type and Inter-Vehicle Distance on Aerodynamic Characteristics during Vehicle Platooning

**Wootaek Kim [1]**, **Jongchan Noh [2]** and **Jinwook Lee [3],***

[1] Department of Mechanical Engineering, Graduate School, Soongsil University, Seoul 06978, Korea; winni4445@gmail.com
[2] Department of Mechanical Engineering, Undergraduate School, Soongsil University, Seoul 06978, Korea; shwhdcks001@naver.com
[3] Department of Mechanical Engineering, Soongsil University, Seoul 06978, Korea
* Correspondence: immanuel@ssu.ac.kr; Tel.: +82-2-820-0929

**Abstract:** Considering the future development in vehicle platooning technology and the multiple models pertaining to complex road environments involving freight cars and general vehicles, the speed and distance of a vehicle model were set as variables in this study. This study aimed at analyzing the effect of currents acting differently using SolidWorks Flow Simulation tool for the vehicle platooning between different models of trucks that are currently being studied actively and sports utility vehicle (SUV) whose market share has been increasing, in order to evaluate the changes in the drag coefficient and their causes. Additionally, purpose-based vehicle (PBV) presented by Hyundai Motor (Ulsan, Korea) during the CES 2020 was considered. In this study, we found that the shape of the rear side of the leading vehicle reduces the drag coefficient of the following vehicle by washing the wake, similar to a spoiler at the rear. The rear side area of the leading vehicle forms a wide range of low pressures, which increases the drag coefficient effect of the following vehicle. The overall height of the leading vehicle also generates a distribution of low pressures above the rear of the vehicle. This reduces the impact of low pressures on the overall height of the following vehicle. The shape of the front of the following vehicle enables the wake of the leading vehicle, which involves low pressures, to inhibit the Bernoulli effect of the following vehicle. Furthermore, the front of the following vehicle continues to be affected by the wake of the leading vehicle, resulting in an increase in the drag coefficient reduction.

**Keywords:** vehicle platooning; drag coefficient; upwash; wake; SolidWorks Flow Simulation

## 1. Introduction

Autonomous driving is a key technology for achieving future mobility. Autonomous driving refers to the technology that enables a vehicle to operate on its own, even without direct inputs from the driver. Rapid advances in data processing and communication techniques, facilitated by the fourth industrial revolution, have improved the applicability of autonomous driving technology. Currently, many global automobile companies are attempting to accelerate the commercialization of autonomous vehicles. Apart from ensuring driver convenience, autonomous driving should reduce unnecessary power consumption by regulating vehicle operation based on the traffic environment, thus, it improves the energy efficiency of the vehicle.

Medium to large-sized commercial vehicle manufacturers use "platooning" to reduce the front resistance of the following vehicle, which is generated due to the eddy currents generated at the rear side of the leading vehicle. Furthermore, significant efforts have been devoted toward developing related technologies to improve the fuel economy by reducing the induced pressure resistance of the leading vehicle by under high pressures, thereby reducing the rear side vortex of the leading vehicle [1,2].

In vehicle platooning, vehicles are driven with short distances between them by using "vehicle gap control" technology. Therefore, the characteristics of platooning with respect to the separation between vehicles should be studied. However, in the study of platooning technology, studies on various vehicle types have not been conducted much, except for the algebraic study of vehicles and simple vehicle speed analysis.

Zabat et al. [3] studied the effect of reducing the average air resistance of clustered vehicles depending on the distance between vehicles during platooning and the number of vehicles and they estimated that the resistance of four vehicles in a cluster was approximately 55% of the resistance of a single vehicle. Tsugawa et al. [4] undertook measurements on test tracks and highways and demonstrated a reduction of 14% in fuel consumption when three trucks were clustered at intervals of 10 m. Jang et al. [5] reported an air force improvement of approximately 7% depending on the reduction in the induced pressure resistance at the rear for different rear structures of the vehicles. Kim et al. [6] reported that the drag coefficient of the leading vehicle varied from $-2.6\%$ to $3.4\%$ when the separation between vehicles was 5–30 m and the same type of vehicle was used. The drag coefficient for following vehicles varied from $-49.1\%$ to $-25.4\%$, as compared to that for a single vehicle.

These prior studies show that the drag coefficient increases with the separation distance and that the drag coefficient increases continuously on moving away from the eddy currents of the leading vehicle. Consequently, realizing platooning with appropriate separation between vehicles can reduce the overall drag coefficient, thereby enabling more efficient operation than that when using individual vehicles.

Therefore, in this study, considering the future developments in platooning technology and the different models, we analyzed the eddy current impact of platooning force using SolidWorks Flow Simulation (version 2015, Dassault Systems, Waltham, France) and the changes in the drag coefficient.

## 2. Research Method

Considering the complex conditions of roads for accommodating autonomous vehicles, four models were built to derive and analyze the changes in the drag coefficient with respect to the distance between two vehicles during vehicle platooning.

### 2.1. Selection and Dimensions of Vehicle Models

The vehicle models selected for this study were as follows: small sport utility vehicle (SUV), medium SUV, purpose-based vehicle (PBV), and trucks (Table 1). For each vehicle, a scenario with the speed recommended by Korean law [7,8] was considered. The small SUV, was modeled based on the vehicle dimensions recommended by the Society of Automotive Engineers (SAE) [9]. The medium SUV was modeled on the medium-sized SUV currently marketed by Hyundai Motor Co. (Ulsan, Korea) [10]. For both small and medium SUVs, motorways suitable for passenger cars were selected as the driving environments; a speed of 110 km/h, which is the maximum legal speed as per highway regulations, was adopted.

The PBV was introduced by Hyundai Motor Co. at CES 2020, and it aimed at wireless charging and platooning while driving with electric vehicles. The vehicle was modeled such that its body was capable of varying from 4 to 6 m in length. An urban speed limit of 50 km/h was selected for the PBV. Furthermore, the developments in platooning technology worldwide were based on the Xiant model [10], which was first applied to trucks and was recently tested for platooning in modern cars; a speed of 90 km/h, the speed limit for highways, was chosen.

**Table 1.** Specifications of the vehicle models used in this study.

| Small SUV | Medium SUV [10] | PBV | Truck [10] |
|---|---|---|---|
| **Side-sectional view** | | | |
| (a) | (b) | (c) | (d) |
| **Top-sectional view** | | | |
| (a′) | (b′) | (c′) | (d′) |
| **Full Length: 3900 mm** | Full Length: 4557 mm | Full Length: 4683 mm | Full Length: 8830 mm |
| **Full Width: 1600 mm** | Full Width: 1900 mm | Full Width: 2300 mm | Full Width: 2490 mm |
| **Full Height: 1200 mm** | Full Height: 1700 mm | Full Height: 2320 mm | Full Height: 3370 mm |
| **Projected area: 1,901,000 mm$^2$** | Projected area: 3,211,000 mm$^2$ | Projected area: 4,845,000 mm$^2$ | Projected area: 8,383,000 mm$^2$ |

*2.2. Numerical Analysis Techniques and Conditions*

2.2.1. Numerical Analysis

A powerful analysis program "SolidWorks" was used to calculate air resistance by employing computerized fluid dynamics (CFD) simulations. The software was used to present the local characteristics. Moreover, as an analysis based on several variables is required, a simplified SolidWorks Flow Simulation model was used to reduce the computation time. The numerical model considered is based on the resolution of the Navier–Stokes equations in conjunction with the standard k-$\varepsilon$ turbulence model [11]. To determine the convergence of the flow analysis, calculations were repeated until the residual fraction of each variable was less than $10^{-4}$%. For higher-fidelity results, more advanced modeling techniques and turbulence models need to be used. These equations were solved via a finite volume discretization method. A constant speed distribution corresponding to the vehicle speed for each model was established under the entrance conditions, and different types of vehicle models with lower speeds were considered for driving in platooning. The exit condition was set the atmospheric pressure. To consider the effectiveness of the moving ground, a parallel velocity component was applied to the ground, and sufficient computational area was provided to the rear of the vehicle model to ensure adequate flow-field development within the test domain. Symmetric conditions were applied to the other walls. In addition, to simplify the analytical boundary conditions, the direction of initial wind remained unchanged; the flow fields of air were defined as follows:

- Turbulent flow;
- Incompressible flow;
- 3-D Steady flow;
- Isothermal flow.

The driving speed of the vehicle model was set to 110 km/h or less, and the 3D Navier–Stokes control equation was solved for the analyzing the incompressible turbulent flow field within the control volume, assuming an incompressible flow. The standard k-$\varepsilon$ model was used for the turbulence analysis.

<Governing Equations>

Continuity equation

$$\frac{\partial U_i}{\partial x_i} + \frac{\partial U_j}{\partial y_j} + \frac{\partial U_k}{\partial z_k} = 0 \tag{1}$$

Momentum equation

$$\frac{\partial U_i}{\partial t} + \frac{\partial}{\partial x_j}(U_i U_j) = -\frac{1}{\rho}\frac{\partial P}{\partial x_i} + \frac{\partial}{\partial x_j}\left[v\left(\frac{\partial U_i}{\partial x_j} + \frac{\partial U_j}{\partial x_i}\right) - \overline{u_i u_j}\right] - g_i \tag{2}$$

Turbulent kinetic energy

$$\frac{\partial}{\partial x_i}(U_j k) = \frac{\partial}{x_i}\left[\left(v + \frac{v_t}{\sigma_k}\right)\frac{\partial k}{\partial x_i}\right] + G - \varepsilon \tag{3}$$

### 2.2.2. Boundary and Initial Conditions

The SIMPLE (Semi-Implicit Method for Pressure Linked Equations) algorithm [11] was applied to calculate the velocity and pressure values in the flow field, and the hybrid scheme [11] was used to calculate the convection term of the standard turbulent model. Moreover, the drag was calculated based on the drag coefficient using Equation (4):

$$C_d = \frac{2\,F_d}{\rho u^2 A} \tag{4}$$

$$\rho = 1.204 \text{ kg/m}^3(Air20\,^\circ\text{C}, 1atm)$$

where $u$: flow speed, $A$: projection area for flow direction.

Boundaries and initial conditions were set as follows:

In case of $k$-$\varepsilon$ model, boundary conditions for $k$-$\varepsilon$ are required. The boundary condition for $k$ is determined from the equation below.

$$k = \frac{3}{2}\left(u_{avg}I\right)^2 \tag{5}$$

where $I\left(= \frac{u}{u_{avg}} \cong 0.16\,Re_{D_H}{}^{-\frac{1}{8}}\right)$ is the turbulence intensity.

$$\varepsilon = C_\mu^{\frac{3}{4}}\frac{k^{\frac{3}{2}}}{l} \tag{6}$$

where $C_\mu$ is the experimental constant ($\approx 0.09$) [12] and $l$ is the turbulence length measure.

- Inlet: speed boundary condition (($V\_SUV = 100$ km/h, $V\_Truck = 90$ km/h, $V\_PBV = 50$ km/h)).
- Outlet: pressure boundary conditions under the condition that the flow field is fully developed (= $1atm$).
- Surface of model vehicle: no-slip boundary condition.
- Low wall: speed boundary conditions equal to vehicle speed.

### 2.2.3. Computational Area Generation Method

Using the general-purpose 3D modeling program "SolidWorks Flow Simulation", a 3D model of the vehicle was created; this file was imported into the numerical analysis domain, and a square grid was created on a rectangular coordinate system. Thus, the grid file for analysis was completed. The optimum grid size of the 3-D model was found by 24 by 16 by 68 from the validation test of numerical grid. The computational domain had the volume and shape of a numerical operational section, as shown in Figure 1. For a single vehicle model, along the $X$-axis, the sides of the domain were at a distance of 8 m from the front of the vehicle and 13 m from the rear of the vehicle; along the $Z$-axis, the left and right sides were both at a distance of 12 m from the center of gravity of the vehicle. Furthermore, along the $Y$-axis, the distance between the lower side of the domain and the underside of the vehicle was 0.2 m. The total length was set to 7 m. For the platooning model, along the $X$-axis, the domain's sides were 8 m from the front of the leading vehicle and 13 m from the rear of the following vehicle. Furthermore, along the $Z$-axis, the left to right in sides of

the domain were both at a distance of 12 m from the center of gravity of the vehicle. Along the *Y*-axis, distance between the lower side of the domain and the underside of the vehicle was 0.2 m away. The total length was set to 7 m.

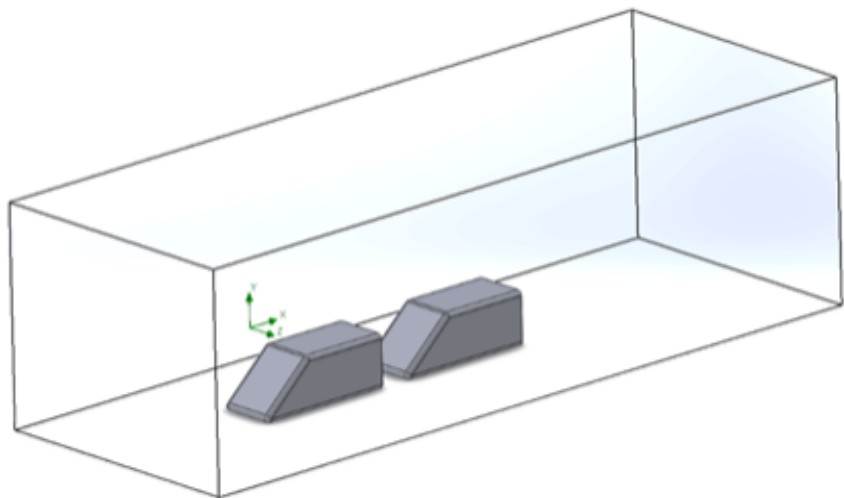

**Figure 1.** Numerical operational section.

2.2.4. Key Variables and Scope for Analytical Research

To evaluate the air force characteristics of the vehicle's driving conditions during platooning, the type of vehicle (i.e., small SUV, medium SUV, PBV, and truck), separation distance (D), and driving speed (V) were set as variables; these are listed in Table 2.

**Table 2.** Analytic conditions.

| Type | Remarks |
|------|---------|
| **Small SUV** **Medium SUV** **PBV** **Truck** | 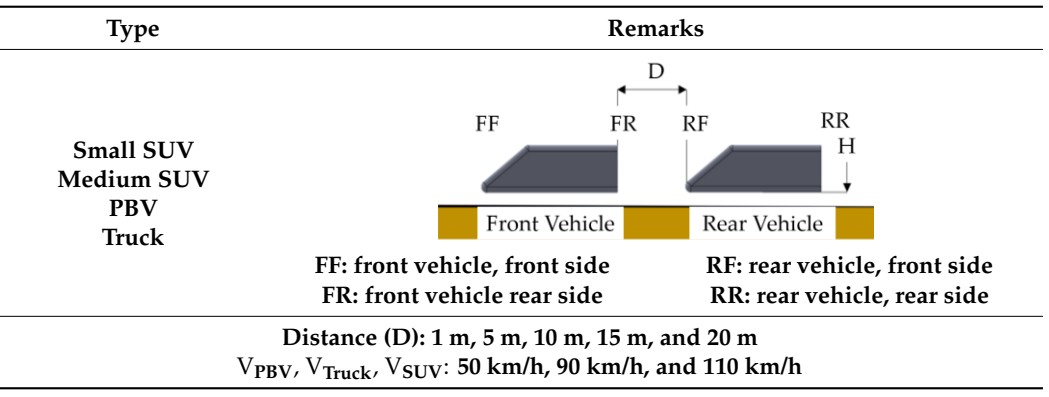 |
| | **FF: front vehicle, front side** **FR: front vehicle rear side** **RF: rear vehicle, front side** **RR: rear vehicle, rear side** |
| **Distance (D): 1 m, 5 m, 10 m, 15 m, and 20 m** **$V_{PBV}$, $V_{Truck}$, $V_{SUV}$: 50 km/h, 90 km/h, and 110 km/h** | |

## 3. Result and Discussion

Numerical analysis was performed based on each following vehicle using the distance between the leading vehicle and separation as variables. Platooning speed was set based on the lower speed of the driving environment involving the leading and following vehicles. The results in terms of distances should scientifically be normalized for reference purposes, in a similar fashion to the drag force. However, this study chose to keep the current format for the following three reasons:

- Ambiguity of normalization target.

This study interprets rear and front features of trailing vehicles as variables, and since the variables affect vary in distance, curvature, width, and height of the features, there is an ambiguous problem in selecting criteria for normalization.

- Legal system aspects.

According to Article 19 (1) of the Korean Road Traffic Act [7], vehicles are obliged to secure safe distances. Therefore, it is currently illegal to maintain close intervals, such as cluster driving, and the legislation for introducing the technology determined that the separation distance (m) based on the inter-vehicle communication-control time would be the standard.

- Industrial application aspects.

This paper is meaningful in demonstrating that the vehicle separation distance and the degree of drag factor reduction are not unconditionally linear. After the autonomous driving was applied, the control standard was also plotted based on the separation distance (m) from the point of view that the trailing vehicle itself would find the optimal resistance point based on data on the rear shape, shape, or rear current of the preceding vehicle.

### 3.1. Variation in Drag Coefficient with Separation Distance

3.1.1. Small SUV as the Following Vehicle

As shown in Figure 2a, the fluctuations in the drag coefficient are consistent, without being considerably affected by the type of leading vehicle. When the separation distance is 1 m, the drag coefficient undergoes the most significant reduction; as the separation distance increases, the drag coefficient also increases. Furthermore, the drag coefficients under separation distances of 15 m and 20 m do not change significantly. This is likely because the overall height of the small SUV is less than those of the other leading vehicles; hence, it is not affected directly by the wake at separation distances of 15 m or more.

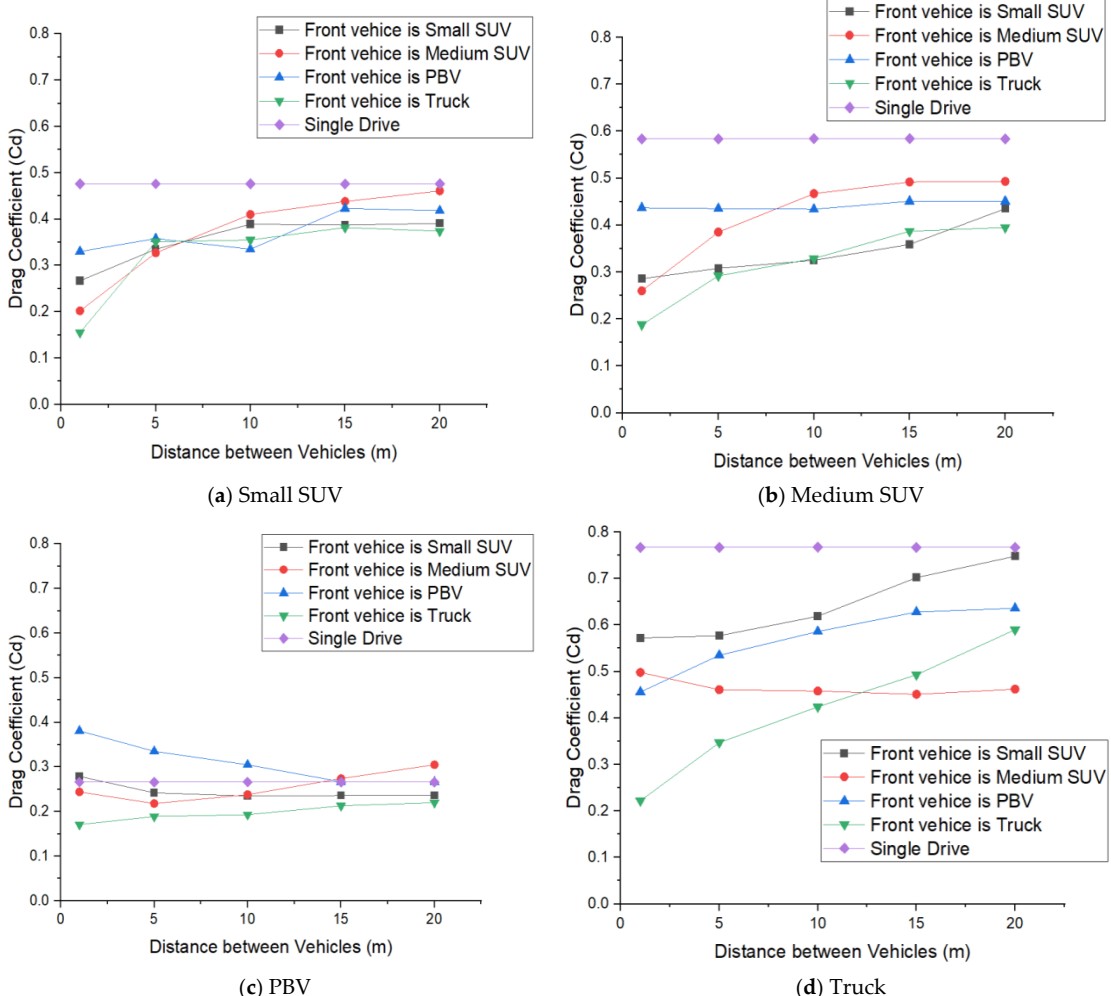

**Figure 2.** Variation in drag coefficient with respect to the leading (front) vehicle type when the following (rear) vehicle is the (**a**) small sport utility vehicle (SUV), (**b**) medium SUV, (**c**) purpose-based vehicle (PBV), and (**d**) truck.

### 3.1.2. Medium SUV as the Following Vehicle

As shown in Figure 2b, similar trends were observed for the drag coefficient, except when using the PBV as the leading vehicle. In these cases, such as when the leading vehicle is the small SUV, the change in the drag coefficient under separation distances of 15 m and 20 m was small owing to the effect of the intensification of the wake of the leading vehicle.

### 3.1.3. PBV as the Following Vehicle

As shown in Figure 2c, compared to that for other vehicles, the reduction in the drag coefficient when using PBV as the following vehicle is not significant. However, owing to the curved surface of the PBV, the resulting stationary pressure at the front of the PBV is considerably low. A reduction in the separation distance results in a decrease in the air flow at the front of the PBV, thereby increasing the drag coefficient. In particular, the most significant increase in the drag coefficient under platooning was observed when the leading vehicle was the PBV. Furthermore, when the leading vehicle is the truck, the drag coefficient remains low owing to the low pressure generated in the wake, which was less than those for the other vehicles.

### 3.1.4. Truck as the Following Vehicle

As shown in Figure 2d, there was no significant change in the drag coefficient, except when the leading vehicle was also a truck with a separation distance of 15 m. When the leading vehicle was the medium SUV, the drag coefficient exceeded that when the leading vehicle was the truck. It has affected the spoiler at the front of the truck by increasing the upwash from the rear of the vehicle, provided that the separation distance increased. Consequently, the intensification of the wake due to the upwash improved the effect of the spoiler, thereby reducing the drag coefficient.

The wake of the PBV reduces drag coefficient due to the diffusion of low pressures; however, it was not affected significantly by the large projected front area of the truck.

### 3.2. *Analysis of the Effect of Drag Coefficient on Separation Distance and Vehicle Type*

For different leading vehicles, the effects of the following vehicles on their drag coefficient with respect to the separation distance was determined.

In general, the larger the area of the leading vehicle, the lower is the drag coefficient of the following vehicle; however, owing to the shape of the rear of the leading vehicle, different trends of the drag coefficient reduction can be observed when using the PBV and medium SUV. As shown in Figure 3c, compared to that when using the PBV alone, the drag coefficient varied by −40.23% during the platooning between PBV and PBV with a separation distance of 1 m. As shown in Figure 3d, at a separation distance of 20 m between the medium SUV and truck, compared to that when using the truck alone, the drag coefficient was 8.5% lower. This resulted in a greater drag coefficient reduction because the rear of the leading vehicle was raised to create an upwash, and the wake acted on the front spoiler of the truck.

Compared to using the small SUV alone, the drag coefficient fluctuates by 67.34% during platooning between the truck and small SUV at a separation distance of 1 m; this is significantly reduced to 26.20% during platooning between the truck and small SUV at a separation distance of 5 m. This also affects the reduction in the drag coefficient of the leading vehicle because a low overall pressure is generated above the rear of the vehicle. The lower the separation distance, the greater the reduction in the drag coefficient. The larger the area of the front of the following vehicle and the greater its overall height, the lower is the pressure generated by the leading vehicle. However, the rear of the leading vehicle either reduces or maximizes the aerodynamic characteristics of the shape of the front, providing variables. The larger the area of the front of the following vehicle and the greater its overall height, the lower the pressure generated by the leading vehicle. However, the rear of the leading vehicle either reduces or maximizes the aerodynamic characteristics

of the shape of the front, providing variables. A summary of the comparisons is presented in Tables 3 and 4.

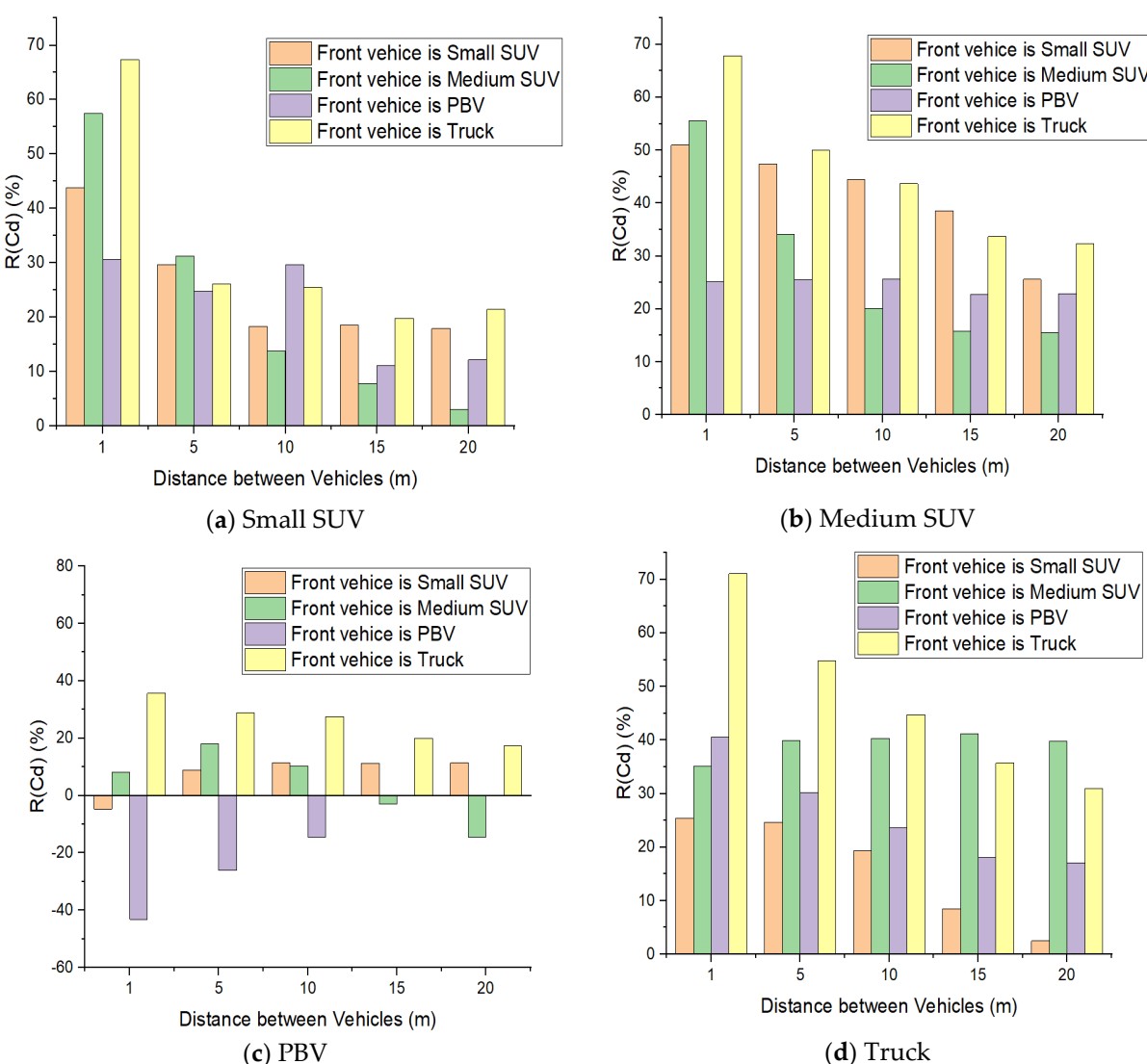

**Figure 3.** Variation in the drag coefficient based on leading (front) vehicle type when the following (rear) vehicle is the (**a**) small SUV, (**b**) medium SUV, (**c**) PBV, and (**d**) truck.

**Table 3.** Results of drag coefficient during platooning.

| Front Vehicle Type | Rear Vehicle (Small SUV) Drag Coefficient ($C_d$) [Single $C_d = 0.476$] | | | | |
| :---: | :---: | :---: | :---: | :---: | :---: |
| | **1 m** | **5 m** | **10 m** | **15 m** | **20 m** |
| **Small SUV** | 0.267 | 0.335 | 0.389 | 0.387 | 0.391 |
| **Medium SUV** | 0.202 | 0.327 | 0.410 | 0.438 | 0.461 |
| **PBV** | 0.330 | 0.358 | 0.335 | 0.423 | 0.418 |
| **Truck** | 0.155 | 0.351 | 0.355 | 0.382 | 0.382 |
| **Front Vehicle Type** | **Rear Vehicle (Medium SUV) Drag Coefficient ($C_d$) [Single $C_d = 0.584$]** | | | | |
| | **1 m** | **5 m** | **10 m** | **15 m** | **20 m** |
| **Small SUV** | 0.286 | 0.308 | 0.325 | 0.359 | 0.435 |
| **Medium SUV** | 0.260 | 0.385 | 0.467 | 0.492 | 0.493 |
| **PBV** | 0.437 | 0.435 | 0.434 | 0.451 | 0.451 |
| **Truck** | 0.188 | 0.292 | 0.329 | 0.387 | 0.395 |

**Table 3.** *Cont.*

| Front Vehicle Type | Rear Vehicle (PBV) Drag Coefficient ($C_d$) [Single $C_d$ = 0.266] | | | | |
|---|---|---|---|---|---|
| | **1 m** | **5 m** | **10 m** | **15 m** | **20 m** |
| **Small SUV** | 0.279 | 0.242 | 0.235 | 0.236 | 0.236 |
| **Medium SUV** | 0.244 | 0.218 | 0.238 | 0.274 | 0.305 |
| **PBV** | 0.381 | 0.335 | 0.305 | 0.267 | 0.267 |
| **Truck** | 0.171 | 0.189 | 0.193 | 0.213 | 0.220 |
| **Front Vehicle Type** | Rear Vehicle (Truck) Drag Coefficient ($C_d$) [Single $C_d$ = 0.767] | | | | |
| | **1 m** | **5 m** | **10 m** | **15 m** | **20 m** |
| **Small SUV** | 0.572 | 0.577 | 0.619 | 0.702 | 0.748 |
| **Medium SUV** | 0.498 | 0.461 | 0.458 | 0.451 | 0.462 |
| **PBV** | 0.456 | 0.535 | 0.586 | 0.628 | 0.636 |
| **Truck** | 0.222 | 0.347 | 0.424 | 0.493 | 0.529 |

**Table 4.** Effect of the shape of vehicles on drag coefficient reduction.

| Factor | | Effect on Drag Coefficient Reduction |
|---|---|---|
| **Front shape of rear vehicle** |  (a) Front element | The high-speed, low-pressure wake of the leading (front) vehicle is suppresses the Bernoulli effect at the front of the following vehicle. |
| **Front area of rear vehicle** | | The large area at the front is continuously influenced by the wake of the leading vehicle, increasing the drag coefficient reduction. |
| **Rear shape of front vehicle** |  (b) Rear element | The drag coefficient reduction of following (rear) vehicles is diminished by washing the wake generated by the rear spoiler, among others. |
| **Rear area of front vehicle** | | The drag coefficient reduction of the following vehicles increases due to the wide, low-pressure range formed at the rear of the leading vehicle. |
| **Overall height of front vehicle** |  (c) height element | The large height creates a distribution of low pressures above the rear, reducing the influence of the low pressures at the rear of the leading vehicle depending on the height of the following vehicle. |
| **Overall height of rear vehicle** | | The small height escapes from the effects of the wake of the leading vehicle more quickly and reduces the effect of the rear vortex as the separation distance increases. |

## 4. Conclusions

In this study, we analyzed the aerodynamic characteristics of platooning using a SolidWorks Flow Simulation for four types of vehicles. By implementing the interaction between the low pressure generated at the rear of the leading vehicle and the high pressure generated at the front of the following vehicle through platooning, the characteristics of the change in the drag coefficient of the following vehicle due to the shape and distance were analyzed.

(1) In most models, the stagnation pressure generated at the front of the following vehicle is suppressed by the low pressure at the rear of the leading vehicle, and the drag coefficient of the following vehicle is generally reduced.

(2)    As small and medium SUVs exhibit similar shapes at the front, their drag reduction trends are similar. When the separation distance exceeds 15 m, the drag coefficient remains constant. Beyond a separation distance of 15 m, the following vehicle is judged to be unaffected by the wake of the leading vehicle.

(3)    When the medium SUV is used, the resulting upwash is the most prominent feature. When the following vehicle is a truck, the rising low-pressure wake at the rear of the medium SUV acts on the spoiler at the top of the truck, thereby reducing the Bernoulli effect; the shorter the separation distance, the higher is the drag coefficient.

(4)    The PBV had the lowest drag coefficient owing to the Bernoulli' effect caused by its outer appearance, and a low pressure was generated near the rear of the PBV. The wake generated at the rear was both raised and washed; as the separation distance increased, it was characterized by the increase of the influence of drag. Therefore, even for a separation distance of 1 m, the reduction in drag coefficient was not greater than that for the other vehicle models. In addition, as the wake of the leading vehicle reduced the Bernoulli effect at the front of the PBV, the shorter the separation distance, the higher is the drag coefficient.

(5)    As the projected area of the truck is greater than those of the other vehicle models, the low pressures at the front and rear have the greatest effect on each other. In addition, the decrease in drag coefficient due to the influence of the spoiler at the front was confirmed; the low pressure above the rear of the truck was generated because the overall height of the following vehicle was low, such as that of a small SUV or medium SUV, for a separation distance of 1 m. Notably, the high pressure at the front of the vehicle did not affect the relief of the low pressure at the rear of the truck.

This study is expected to help control the separation distance to achieve optimal driving efficiency when driving in groups involving various types of vehicles in the future. As a follow-up study, it is necessary to identify the main factors that reduce the drag coefficient during platooning by using the design of experiment (DOE) method and to normalize various models in order to secure reliability through more advanced modeling/turbulence models.

**Author Contributions:** W.K. and J.N. conceived and designed the data analysis, analyzed the data, and wrote the paper; W.K. and J.N. performed the calculation and produced the numerical data; J.L. supervised and advised on all parts of this paper. All authors have read and agreed to the published version of the manuscript.

**Funding:** This research received no external funding.

**Institutional Review Board Statement:** Ethical review and approval are not applicable for this study not involving humans or animals.

**Informed Consent Statement:** Informed consent was obtained from all subjects involved in the study.

**Data Availability Statement:** The data presented in this study are available on request from the corresponding author.

**Acknowledgments:** This research was supported by Soongsil University, Republic of Korea.

**Conflicts of Interest:** The authors declare no conflict of interest.

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
