# Peer review of "Effects of Vehicle Type and Inter-Vehicle Distance on Aerodynamic Characteristics during Vehicle Platooning"

_applsci, doi:10.3390/app11094096_

Round 1

Reviewer 1 Report

The authors conduct a parametric study on a rather unusual and very very interesting subject. There is no doubt that this article would be of interest to the community, however there are some points that need to be addressed. I have some remarks, which are directly related to the scientific soundness of the article (most notably on modeling methods and results normalization) and I consider necessary for journal publication.

Moreover, I have some general comments that will help eliminate skepticism, e.g. related to the rather simple modeling environment.

Remarks:

- Selection and dimension of the models: Please make sure that all of your selections are directly linked with a citation. For example, the authors must explicitly state the sources on which the numbers are taken from, in the caption of Table 1. In case some number/configuration is based on an assumption made by the authors, I strongly advice that it is mentioned in the manuscript.

- Normalization: This is a parametric study and is rather clear to me that the phenomena shown here could potentially affect other types of vehicles as well (e.g. a "super mini" type of vehicle). Hence, I would like the authors to normalize their results. Note that you have already used the normalized Cd coefficient and not the force in Newtons, for the exact same reason.

In other words, to state that the distance between two vehicles is 15m is of very little interest to the academic/industrial community, as it limits your conclusions to your study only (and this is rather unfair for your work!). Please make sure that those distances are normalized using a corresponding value from the vehicles, e.g. distance/vehicle length. That way, the values of Table 3 will appear in relation to the vehicle, and you might as well find some similarity between the different types of vehicles by using normalized values.

- Modeling: I understand that the modeling environment is not tailored to the needs of an aerodynamic analysis. However, I see nothing wrong here (please see my comment #4). What is a bit off, however, is the way that the authors present their governing equations and their boundary conditions.

As a start, unless I am missing something, there is no reason (no compressibility effects, no heat transfer) to use the energy equation (even if you used it and you cannot re-run your simulations, at least do not present it in the manuscript...). Moreover, the presentation of the turbulence terms is very basic, as is the information provided on the control volume. Please find a couple of citations to back-up your selections, otherwise they seem too arbitrary. Last but not least, though, there is no information provided on the computational mesh. Again, looking at a similar study can help the authors to resolve this serious issue (for example, you should mention something about a grid dependency study)

Provided that those, more serious, remarks are addressed, I have some other general (minor) comments:

  1. I would advice that the authors do not use first-person in the manuscript.
  2. Please make sure that the views of the vehicles are annotated, especially at Table 1.
  3. In a similar comment to #2, please find a way of annotating the different vehicles at Table 4 (I am not sure if you have to show them here, though)
  4. Solidworks is a rather simple modeling environment when it comes to fluid dynamics and aerodynamics. Again, you do not have to change your simulations. I advice the authors to write a comment for this issue, e.g. stating that this is a first step of this investigation and that a more advanced modeling technique/turbulence model must be employed for higher-fidelity results.
  5. I would really like to see the boundary conditions in a table (similar to Table 1), also mentioning the corresponding citations.
  6. The projected area should include no more than 4 significant figures, especially since the units are in mm2 (especially decimals should be avoided)
  7. Since this is a parametric study with various parameters, the authors may want to consider a Design of Experiments (DoE) approach for future work, or at lest suggest it in the conclusions. For example, I can see some applicability of the Taguchi method here.

Summing up, I strongly suggest that the authors add some additional effort and address the above points. Again, I really really liked the subject and believe that it is 100% worth of publishing.

Author Response

We are very appreciate for your precious reviewing.

In the revised paper, the English part of the correction is written in blue, and the parts reflecting the main reviewing opinions are marked in red and the answer is attached with PDF file.

Reviewer 2 Report

In the present manuscript, the authors analyze the effect of the currents acting differently through the SolidWorks Flow analysis of platooning between different models to analyze the trend of change in drag coefficient and its cause, along with trucks currently being actively studied, as well as SUVs , which are expanding in market share. However, I will describe some changes that the authors should make to improve the quality of the article:
-The presented format of the article is not updated. In addition, this format is not the one for review, so the lines are not numbered and it is difficult to detail where the errors are for the reviewers.
-The authors use the acronyms incorrectly, the correct way is to capitalize the first letter that describes the acronym, such as "Purpose-Based Vehicle (PBV)". This incorrect way of using acronyms must be corrected in all acronyms of the manuscript.
-The authors have not written some meanings of the acronyms they use, so it may be difficult for the reader to have an easy understanding.
-The authors have not written a Related Works Section. Within this Section they should write a brief presentation of the Sections of the article.
-There is no explanation of the terms of the Equations.
-They incorrectly use the quotation of the Equations in the manuscript.
-There is no explanation of Quasi-3D
-Why is there underlined text after Equation 5?
-It must include an introduction to the Simulator that the authors are using.
-The authors must use the complete word when mentioning "Figure", "Equation", "Table", "Algorithm", "Section", and not sometimes an abbreviation or other times complete.
-Which scenarios have the authors studied? There is no mention.
-Where is the traffic data obtained for the simulation?
-What are the parameters of the simulation? What physical environments are they using?
-More experimentation is needed in different scenarios.

Author Response

(The authors gave the same response as above.)

Round 2

Reviewer 1 Report

The authors have reviewed most of the remarks, however some key points remain unaddressed. The last sentence is like a summary of the points that were not actually addressed.

It is acceptable to state that DoE will be considered for future work.

Normalizing the results and showing more information about the turbulence models is not part of a future work, though - it is a part of this manuscript before publishing.

More specifically:

  1. No citations are shown in the data provided in Table 1
  2. I understand and respect the motivation for future work. If the authors do not normalize the results and present them as distances in meters, the scope of this work is greatly limited. It is like showing the forces in Newtons, instead of normalized coefficients (CD). Please make sure that the results are normalized.
  3. Mesh information needs to be reviewed and polished. I regret to say that the phrase "26,000 grids" is not proper terminology. Moreover, and most importantly, I still see no update on boundary conditions (turbulence intensity or length scale) and no justification for the selection of the turbulence model. The authors must refer to a published high-quality CFD analysis for proper phrasing, so that the scientific soundness is adequate for publishing.

I strongly suggest that some extra effort is added and that those remarks are properly addressed before achieving scientific publication.

Author Response

The revision was made to reflect the review's opinion, and I would like to submit the revised paper. Thanks for your valuable comments with greatest care.

Reviewer 2 Report

The present manuscript has not performed the pertinent changes suggested by the authors, ranging from the correct format of the MDPI to including Works Related to the research performed by the authors.

Author Response

(The authors gave the same response as above.)

Round 3

Reviewer 1 Report

The authors have completed addressing all remarks in the manuscript, apart from Point 2. It has been perfectly addressed to me, but not in the manusript.

The three reasons you mention (ambiguity, legal aspects, industrial aspects) are very good reasons to keep the results as they appear:

But why do you keep those arguments out of the manuscript?

Please write a paragraph at the end of the results section, where you state exactly this, i.e. that

"the results in terms of distances should scientifically be normalized for reference purposes, in a similar fashion to the drag force. However, the authors choose to keep the current format for the following three reasons:

1. (...)

2. (...)

3. (...)

"

Please address this detail before publishing - I will then accept the manuscript.

Author Response

Please confirm the attached file. We'd like to thanks for your valuable comments with greatest care. 
